ecology

boreal ecosystems, climate change, indirect interactions, population cycles, predator–prey interactions

**Author for correspondence:**
Diana E. Bowler
e-mail: diana.e.bowler@gmail.com

# Impacts of predator-mediated interactions along a climatic gradient on the population dynamics of an alpine bird

Diana E. Bowler[1,2,3,4], Mikkel A. J. Kvasnes[4], Hans C. Pedersen[4], Brett K. Sandercock[4] and Erlend B. Nilsen[4,5]

[1]Department of Ecosystem Services, German Centre for Integrative Biodiversity Research (iDiv), Putschstr. 4, 04103 Leipzig, Germany
[2]Institute of Biodiversity, Friedrich Schiller University Jena, Dornburger Straße 159, 07743 Jena, Germany
[3]Department of Ecosystem Services, Helmholtz Center for Environmental Research—UFZ, Permoserstraße 15, 04318 Leipzig, Germany
[4]Department of Terrestrial Biodiversity, Norwegian Institute for Nature Research, P.O. 5685 Torgarden, 7485 Trondheim, Norway
[5]Nord University, Faculty of Biosciences and Aquaculture, Steinkjer, Norway

DEB, 0000-0002-7775-1668; MAJK, 0000-0002-6603-337X; BKS, 0000-0002-9240-0268; EBN, 0000-0002-5119-8331

According to classic theory, species' population dynamics and distributions are less influenced by species interactions under harsh climatic conditions compared to under more benign climatic conditions. In alpine and boreal eco-systems in Fennoscandia, the cyclic dynamics of rodents strongly affect many other species, including ground-nesting birds such as ptarmigan. According to the 'alternative prey hypothesis' (APH), the densities of ground-nesting birds and rodents are positively associated due to predator–prey dynamics and prey-switching. However, it remains unclear how the strength of these predator-mediated interactions change along a climatic harshness gradient in comparison with the effects of climatic variation. We built a hierarchical Bayesian model to estimate the sensitivity of ptarmigan populations to inter-annual variation in climate and rodent occurrence across Norway during 2007–2017. Ptarmigan abundance was positively linked with rodent occurrence, consistent with the APH. Moreover, we found that the link between ptarmigan abundance and rodent dynamics was strongest in colder regions. Our study highlights how species interactions play an important role in population dynamics of species at high latitudes and suggests that they can become even more important in the most climatically harsh regions.

## 1. Introduction

Climatic variability and species interactions are two key drivers influencing the spatial and temporal patterns in the distribution and abundance of organisms [1,2]. An old hypothesis, originally proposed by Darwin, posits that climate is the main determinant of species' range limits in harsh abiotic regions whereas species interactions are a more important determinant under benign abiotic conditions (hereafter the 'classic hypothesis') [3]. Range limits are determined by local population growth rates; hence, this hypothesis also implies that population growth rates should be more sensitive to climate variability in climatically harsh regions but more sensitive to species interactions in more climatically benign regions. Previous studies have examined changes in demographic and population processes along elevational and latitudinal climatic gradients [4–7], but few studies have been able to tease apart the sensitivity of population growth to climate and species interactions over large spatial and temporal scales.

In the cold environments of alpine and boreal areas, vertebrate species coexist in relatively simple food-webs but biotic interactions still exert strong

impacts on species' population dynamics [8]. One particular characteristic of northern alpine and boreal ecosystems in Fennoscandia is the rodent population dynamics, which are characterized by large interannual cyclic fluctuations in abundance, with peak years coming at regular intervals [9,10]. Similar cycles are seen with other species in other parts of the world e.g. the snowshoe hare (*Lepus americanus*) in the boreal ecosystems of North America [11]. Although the mechanisms are still contested, predator–prey interactions are generally regarded as the most parsimonious explanation underlying the cyclic dynamics of rodents [12,13], especially due to specialist predators such as mustelids [14].

The population dynamics of many species of alpine and boreal communities, including the avian community, often covary with the phase of the rodent cycle [15,16]. One hypothesized reason for this covariation is that ground-nesting birds (such as ptarmigan and grouse, Tetraonidae) and rodents share a similar set of generalist predators, including the red fox (*Vulpes vulpes*) and corvids [17,18]. Hagen [19] proposed the 'alternative prey hypothesis' (APH), stating that ground-nesting birds are less vulnerable to predation in peak rodent years because generalist predators switch to feeding on rodents. By contrast, ground-nesting birds suffer greater predation in rodent decline/crash years because there are fewer prey options available to hunting predators. Other indirect interactions based on shared predators, such as apparent competition, are typically predicted to lead to negative correlations in the abundance of alternative prey species [20]. Here, predator-mediated interactions between rodents and ground-nesting birds are hypothesized to lead to a positive association, such that ground-nesting birds are more abundant in peak rodent years. Similar indirect effects of rodent population dynamics have been examined in a range of ecosystems, and across a range of alternative prey species [16,21,22].

Previous studies have found partial support for the main predictions of the APH [16,23]; however, the APH also makes secondary predictions about how the relative importance of predator-mediated interactions should change along ecological gradients. In particular, Angelstam *et al.* [15] predicted that interactions should be stronger in regions with fewer prey species because the predators become functionally more specialist, causing tighter linkages between predators and prey. By contrast, in regions where predators have access to a more diverse array of prey species, fluctuations in rodent populations should have weaker effects on the dynamics of predator populations. Indeed, the reduced amplitude of the rodent population cycle at lower latitudes is probably due to a stabilizing effect of more generalist predators [24–26]. Climatic conditions, especially snow cover, has also been linked with characteristics of the rodent cycles, with more cyclic behaviour in regions with greater snow cover [16,27,28]. Based on these arguments, ground-nesting birds might be less affected by the rodent cycle in less climatically harsh areas, where there is a more diverse set of prey resources and weaker linkage between rodents and predators.

Alpine birds in Fennoscandia are reported to be declining in abundance over the last decades [29], but the underlying causes remain poorly understood. Willow ptarmigan (*Lagopus lagopus*) are a resident species of montane habitats. Previous studies of population dynamics have shown that the abundance of willow ptarmigan typically fluctuate in synchrony with the rodent dynamics [16,30]. In addition, recruitment rates are affected by weather conditions during the breeding season [30]. Thus, ptarmigan are a good model species for testing alternative hypotheses about the relative importance of climate forcing and species interactions on population dynamics. In this study, we compared the sensitivity of the population dynamics of willow ptarmigan to temporal variability in both climate and rodent populations using a multi-year (2007–2017) dataset covering a large geographical gradient across Norway. We assumed no direct effect of rodents on ptarmigan but rather we expected their population dynamics to be linked due to prey-switching of shared predators as predicted by the APH. Therefore, we used the statistical relationship between rodent abundance and ptarmigan population dynamics as a signal of predator-mediated interactions (figure 1). Although we use the term 'harsh' to describe the coldest climatic conditions within our study sites, we note that they are not necessarily harsh for the ptarmigan, a species well-adapted to cold conditions. We tested three key predictions that emerge based on the classic hypothesis about the relative importance of climate versus species interactions, and the APH (figure 1):

(1) Under the classic hypothesis, we predicted that predator-mediated interactions, and hence rodent population dynamics, would more strongly affect ptarmigan in warmer regions. By contrast, under the APH, we predicted that predator-mediated interactions, and hence rodent population dynamics, would more strongly affect ptarmigan in colder regions, where there are fewer prey sources for predators and stronger trophic linkages between predators and their prey populations.

(2) Under both hypotheses, we predicted that ptarmigan population growth would be more sensitive to interannual climatic variability in colder regions, because they are closer to the edge of their physiological (climate) and ecological (food availability) niche, which may render them more sensitive to climatic variability.

(3) Last, we predicted that the standardized effect sizes of rodents on the population growth rate of ptarmigan would be generally greater than the effect of climate because ptarmigan are well-adapted for life in cold environments and predation is a major determinant of reproductive success.

## 2. Material and methods

### (a) Population abundance data

We used a dataset of line-transect surveys covering almost the full latitudinal extent of Norway across an 11-year period (2007–2017). The surveys come from a structured citizen science programme, based on coordinated collection by local and regional initiatives. Volunteer surveyors were instructed to follow common field procedures for estimation of ptarmigan densities using distance-sampling methods [31]. At a national or regional scale, however, the programme does not allow for a random selection of study sites, because site selection is focused on the alpine habitats used by ptarmigan and is contingent on landowners establishing surveys on private and public lands. During the surveys, a team of two persons followed pre-defined transect lines, using trained pointing dogs to search both sides of the transect line, usually during August. For each detection of a cluster of ptarmigan (a group or an individual), surveyors recorded the perpendicular distances of the ptarmigan from the transect, the number of birds, the geographical coordinates and the date and time of day. In addition, the length (and geographical position) of the transect line is

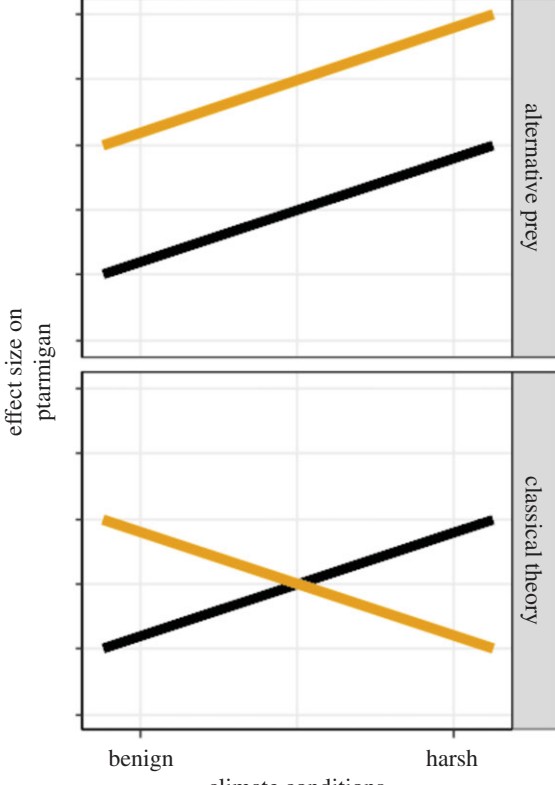

**Figure 1.** (*a*) The hypothesized direct (solid) and indirect (dashed) relationships between ptarmigan and rodents as alternative prey, and their generalist predators, such as the red fox. In peak rodent years, ptarmigan are mostly ignored by predators; by contrast, in crash rodent years, ptarmigan are more frequently depredated. Together, these processes lead to a positive effect of rodent abundance on ptarmigan abundance. (*b*) Predicted effects of interannual climatic variability (black lines) and rodent abundance (orange lines) on ptarmigan population dynamics according to the classic hypothesis (bottom) versus the 'alternative prey hypothesis' (top). (Online version in colour.)

recorded, as well as some other auxiliary data [32]. Since 2013, all data management, development of field procedures and planning of study design has been coordinated by the Hønsefuglportalen project (http://honsefugl.nina.no) in close collaboration with participating landowners. Transects are spatially clustered into survey regions (median of 17 line-transects per survey region), reflecting data that are reported to the same survey coordination node. We used all data collected since 2007, when the monitoring was substantially expanded, until 2017. The total effort varied across the period, and not all study sites were surveyed each year. We filtered the dataset to 708 line-transects that were visited in at least 6 years (a

median of 10 years), each with a mean length of 3.7 km. We excluded observations made at distances greater than 200 m from the transect line, as well as detections by the surveyor, and not by the dog, at distances greater than 10 m away from the transect line because those observations were likely due to the surveyor flushing birds when moving towards the pointing dog which were observed only because the surveyor left the transect line.

## (b) Covariate data

### (i) Rodent data

During the line surveys, surveyors recorded whether rodents were observed along the transect as a binary variable (yes/no). The surveyors were not asked to distinguish among rodent species, but the five main species in the study area that show cyclic dynamics include: Norwegian lemming (*Lemmus lemmus*), tundra vole (*Microtus oeconomus*), field vole (*M. agrestis*), bank vole (*Myodes glareolus*) and the grey red-backed vole (*M. rufocanus*). Rodent occurrence is denoted by *R* in subsequent statistical models.

### (ii) Climatic data

We obtained high-resolution climatic data from the Norwegian Meteorological Institute (MET Norway: www.met.no). Data were based on an interpolation from local weather stations, with a spatial resolution of $1 \times 1$ km and are available on a daily basis. For each line-transect, we applied a buffer of 200 m and matched our transect locations with the climate data. With the compiled data, we characterized the spatio-temporal variation in climatic conditions during spring (prior to and at the start of the breeding period) and winter (marking the end of the breeding period). Specifically, we calculated:

(1) Spring onset—first year day with a rolling 7-day mean snow depth of zero. We use the term 'spring' loosely since snow cover did not completely disappear until summer in some regions.
(2) Spring temperature—average daily mean temperature during March, April and May.
(3) Winter onset—first year day with a rolling 7-day mean snow depth above zero.
(4) Winter temperatures—average daily mean temperature during December, January and February.

For each climatic variable, we calculated (1) the average climatic conditions for each transect between 2007 and 2017, which defined the spatial climatic harshness gradient (denoted as CS) and (2) the difference between the climatic value of each year at each transect and the spatial climatic variable of each transect, which defined the temporal variation (i.e. anomalies, denoted as CT). Hence, we considered the effects of different characterizations of the climatic harshness gradient.

### (iii) Harvesting data

We also compiled available data on harvest bags on ptarmigan but these data were only available at a coarse scale of administrative units and not at the line-transect scale, and thus enables only a weak test of the effects of harvest (see electronic supplementary material, A for further details—no effect of harvesting was found in our subsequent analysis on the resolution analysed here, but the covariate was retained in our model regardless).

## (c) Statistical analysis

### (i) Ptarmigan dynamics

We modelled ptarmigan dynamics using a hierarchical Bayesian model that included two linked models: a model for the observation process along each transect line (i.e. the observation

model) and a model that described the ecological dynamics including the effect of climate and rodents on ptarmigan density as the latent variable (i.e. the state model).

(1) *Observation model: estimation of effective strip width of the ptarmigan line-transects.* We fitted a distance-sampling detection model to estimate the effective strip width (ESW) of each transect [31]. The central assumption of the distance-sampling theory is that the detection probability decreases with increasing distance from the line-transect. We modelled the perpendicular distances of observation events (one or more birds in a group) from the transect as a half-normal distribution. On the transect line, we assumed perfect detection—a common assumption in distance-sampling [31]. We modelled sigma—the parameter of this half-normal distribution that reflects the rate of distance-decay of detections—to be dependent on group size (i.e. the number of birds in each observation). Mean group size was modelled with a Poisson mixed-effects model with line and survey region, interacting with year (as a factor). To allow for any spatial variation in sigma, we additionally included random effects for transect line ($L$, in the equation below) and survey region (SR), allowing some differences in how detectability declines with distance due to landscape or habitat features. We also considered fixed effects of forest cover or precipitation during the survey but found no effects (95% credible intervals overlapped zero) and thus did not include these factors in the model.

The final model for sigma was

$$\log(\text{sigma}_{i,t}) = b_0 + b_{CS}\text{Group size}_{i,t} + L_i + \text{SR}_i.$$

The ESW of each transect ($i$) in each year ($t$) was calculated from sigma based on the following equation:

$$\text{ESW}_{i,t} = \sqrt{\frac{\left(\pi * \text{sigma}_{i,t}^2\right)}{2}}.$$

(2) *State model: testing the effect of climate and rodent on ptarmigan populations.* We then used the estimated ESW and data on transect length (TL) to relate the total number of individuals observed along each transect ($N$, as following a negative binomial distribution with dispersion parameter, $r$) to the latent variable, ptarmigan density, $D$, (abundance per km$^2$) for each year $t$ and each transect $i$:

$$\text{Ptarmigan\_Obs}_{i,t} \sim \text{Negative Binomial}(N_{i,t}, r)$$

and

$$N_{i,t} = D_{i,t} \times \text{TL}_{i,t} \times \text{ESW}_{i,t} \times 2.$$

Ptarmigan density ($D_{i,t}$) was modelled in two different ways. In the first model (random-effects model), we modelled the ptarmigan dynamics using a series of random effects that reflected the spatial and temporal structure of the data. The random terms included year (as a factor), transect line, survey region and survey region-year. We used this model to visualize of the realized ptarmigan dynamics, without explicitly specifying the underlying ecological covariates.

In the second set of models (mixed-effects model), we tested the additive effects of the two main ecological variables of interest: rodent occurrence and climate as well as the interaction between rodent occurrence and the spatial climatic variable (table 1). We tested the four climatic variables (winter temperature, winter onset, spring temperature, spring onset) in separate models.

Density in year 1 was modelled using spatial random effects among line-transects. Density in subsequent years was modelled as spatial ($i$) and temporal ($t$) variation in the population growth rate as

$$\ln D_{i,t} = \ln D_{i,t-1} + r_{i,t},$$

**Table 1.** Table of model parameters on ptarmigan density and their meanings.

| term of model | type | statistical meaning |
| --- | --- | --- |
| $b_0$ | fixed | intercept |
| $b_A$ | fixed | density-dependence |
| $b_{CS}$ | fixed | spatial climatic variable |
| $b_{CT}$ | fixed | temporal climatic anomalies |
| $b_R$ | fixed | rodent occurrence (same year) |
| $b_{RL}$ | fixed | lagged effect of rodent occurrence (previous year) |
| $b_H$ | fixed | harvesting during the previous hunting season |
| $b_{\text{int\_CT}}$ | fixed | interaction term between temporal and spatial climate |
| $b_{\text{int\_R}}$ | fixed | interaction term between rodents (same year) and spatial climate |
| $L$ | random | line-transect effect |
| SR | random | survey region effect |
| year | random | year effect |

where the growth rate, $r$, was decomposed into (table 1):

$$\begin{aligned} r_{i,t} = {}& b_0 + b_A \ln D_{i,t-1} + L_i + \text{SR}_i + \text{Year}_t + b_{CS}\,\text{CS}_i + b_{CT}\,\text{CT}_{i,t} \\ & + b_R\,R_{i,t} + b_{RL}\,R_{i,t-1} + b_h\,H_{i,t-1} + b_{\text{int\_CT}}\,\text{CS}_i\,\text{CT}_{i,t} \\ & + b_{\text{int\_R}}\,\text{CS}_i\,R_{i,t}. \end{aligned}$$

where $b_A$ assumes a Gompertz form of density-dependence, an assumption made by previous ptarmigan studies [33,34]. $L$ and SR were random effects to account for the spatial grouping of data into line-transects ($L$) and survey region (SR). A year (as a factor) random effect was included to account for any additional causes of temporal variability.

CS and CT were covariates for the spatial gradient in climate and temporal anomalies for the climatic variable, respectively. Two coefficients ($b_R$ and $b_{RL}$) for rodents ($R$) tested the effects of rodent occurrence immediately preceding the ptarmigan survey (during the same year, $t$) and a lagged effect of rodent occurrence during the previous year (RL, $t-1$). We also accounted for the potential effect of harvesting ($H$) on ptarmigan growth. All covariates were centred and scaled to units of 1 standard deviation prior to analysis.

The critical tests of the classic versus the alternative prey hypotheses were based on the interaction terms of our model. The interaction terms were: (i) between CS and CT to test whether the effect of annual climatic anomalies changed along the climate gradient or (ii) between CS and $R$ (of the same year) to test whether the effect of rodents changed along the climatic gradient.

In preliminary analyses, we explored additional random effects (a coarser spatial term) but they had little effect on our results so we retained the simpler model.

## (ii) Rodent dynamics

We included a submodel for rodent occurrence (presence/absence, modelled as a Bernoulli distribution) within our hierarchical model for ptarmigan. This submodel was based on a series of random intercept terms (for survey region [SR], year [$Y$], as a factor and their interaction) to flexibly predict the interannual variability in

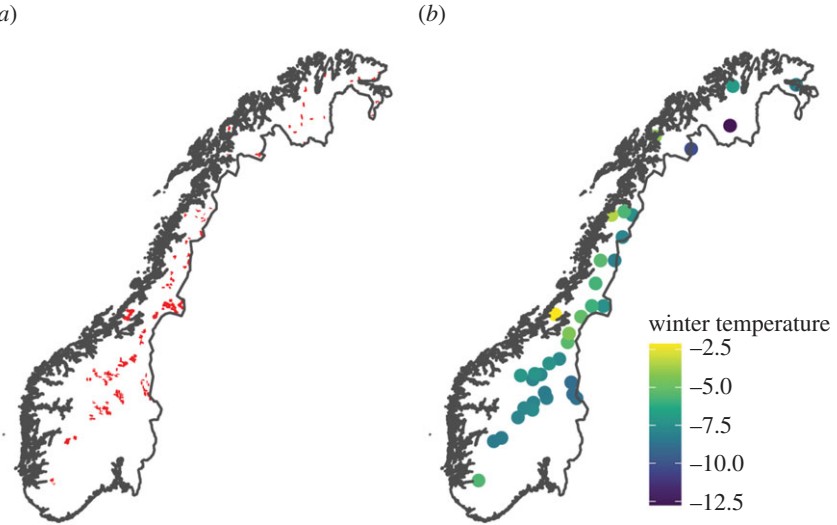

**Figure 2.** (a) Location of the 708 line-transects across Norway are shown by red lines. (b) The location of each of the 36 survey regions (a survey region is a cluster of neighbouring line-transects), coloured by their mean winter temperature during 2007–2017. Electronic supplementary material, figure S1 shows the same maps for the other climatic variables. (Online version in colour.)

rodent dynamics. Hence, the rodent submodel was

$$\text{Rodent\_Occ}_{i,t} \sim \text{Bernoulli}(R_{i,t})$$

and

$$\text{logit}(R_{i,t}) = b_0 + Y_t + \text{SR}_i + Y_t\,\text{SR}_i.$$

Modelling rodent occurrence allowed us to input missing data in the observed rodent dataset. Additionally, the model predicted the probability of rodent occurrence at the level of the survey region rather than at the transect-level, which meant smoothing over local reporting and ecological variation. This choice was based on the assumptions that many predators (especially birds of prey) forage over reasonable distances and are not likely affected by fine-scale transect-level variation. We also considered a more complicated autoregressive model of order 2 for the rodent dynamics, but similar cyclic dynamics were obtained, and similar effects on ptarmigan were found (electronic supplementary material, B), so we proceeded with the simpler assumption-free random-effects model.

In further models, we also investigated how rodent dynamics changed along the climatic gradient. In Bayesian mixed-effects models with binomial errors, we tested the effects of the spatial climatic variables and temporal climatic anomalies variables on the probability of rodent occurrence. Line-transect and survey regions were also included as random intercepts. Each of the four types of climatic variables (winter temperature, winter onset, spring temperature, spring onset) were tested in separate models.

Models were fitted using JAGS within R v. 3.6.0. We used three MCMC chains, ran for 50 000 iterations, discarding the first 25 000 iterations as a burn-in period. We used uninformative or weakly informative priors for all parameters, but a narrower prior for the density-dependence term (a uniform prior between −2 and 0) to help convergence. We assessed model convergence using MCMC chain traceplots and the Gelman–Rubin R-hat statistic (all less than 1.1). To assess the fit of the model, we used the DHARMa package [35] that uses simulations from the fitted model to create scaled residuals, which were then checked against the expected distribution using a quantile–quantile plot and against the model predictions to check for any systematic deviations in fit. Covariate effects were inferred when the 95% credible intervals did not overlap zero. Code for the fitted model is available in the electronic supplementary material, C.

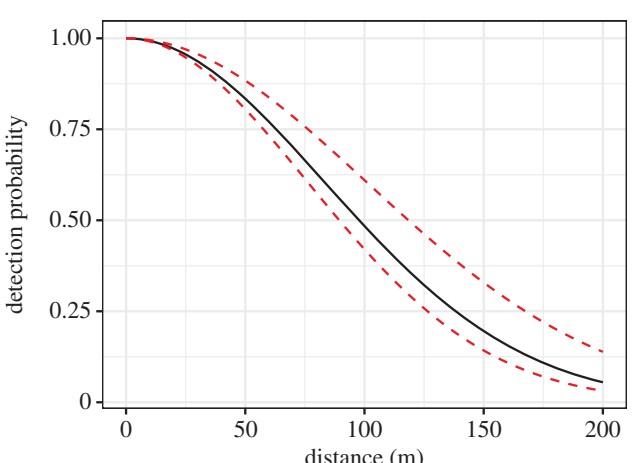

**Figure 3.** Distance-dependent decay of ptarmigan detectability along the line-transects. The black line is the prediction for the median ptarmigan group size; the lower and upper dashed red lines are the predictions for the minimum and maximum ptarmigan group size. (Online version in colour.)

## 3. Results

### (a) Data description

The 708 line-transects (figure 2*a*) were clustered into 36 survey regions (figure 2*b*) distributed along a wide latitudinal and climatic gradient across Norway. The warmest regions were those nearest the coast and the coldest regions were inland and upland (figure 2*b*). Along each transect, a median of two groups were observed (range = 0–18), with a median of four (1–35) individuals seen per group. As expected by distance-sampling theory, the frequency of observations declined with increasing distance from the line-transect (figure 3). The average ESW was 107 m (interquartile range = 100–114 m). Sigma of the half-normal distribution was positively affected by group size, with the estimated ESW being 95 m for the smallest group size and 126 m for the largest group size (figure 3).

### (b) Climatic, rodent and ptarmigan population variation

Climate and rodent occurrence both showed large and generally spatially synchronized fluctuations (figure 4, note:

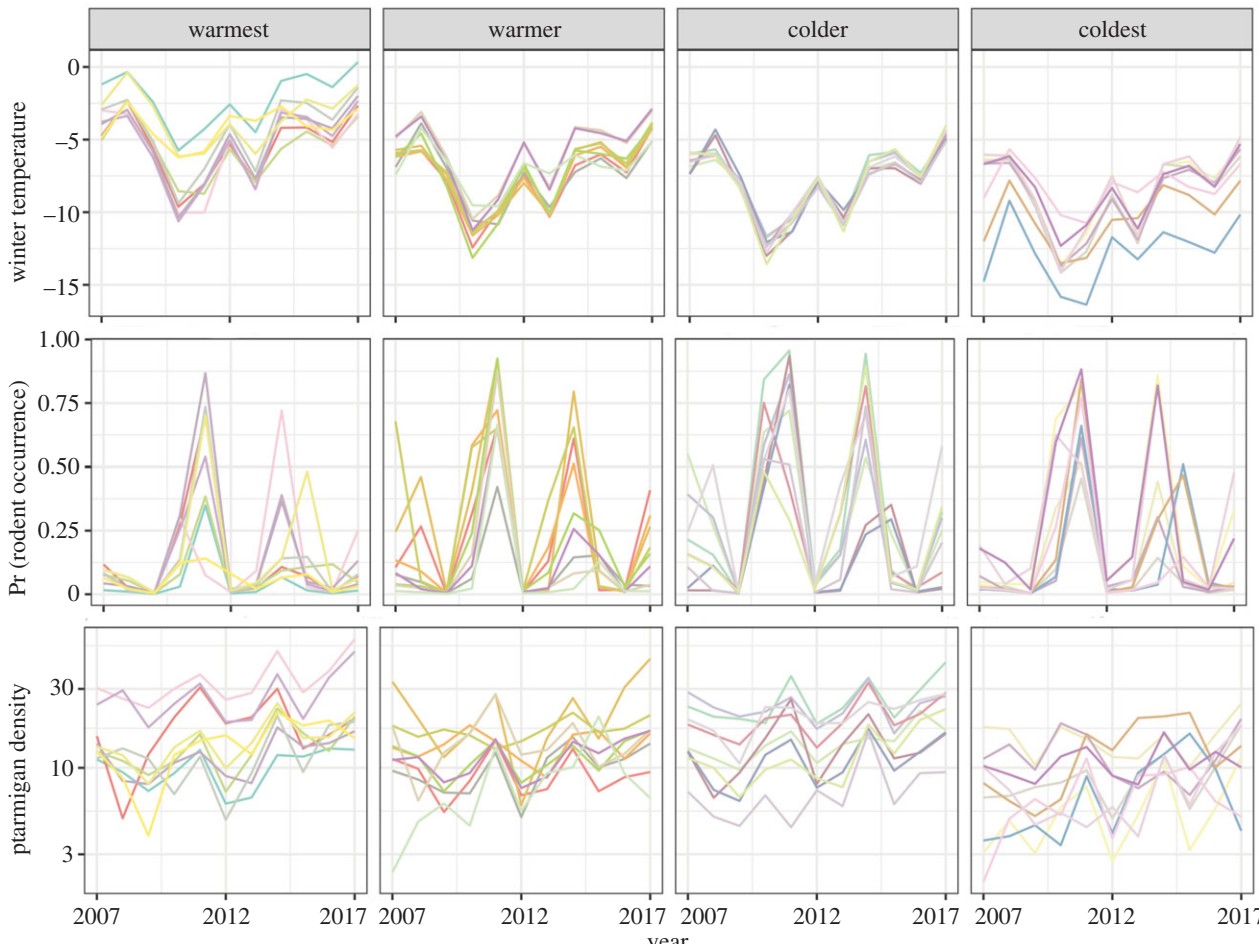

**Figure 4.** Time-series of winter temperature (°C), probabilities of rodent occurrence and willow ptarmigan densities in August (birds per km²) as predicted by the random-effects models. In our analysis, the spatial climatic gradient was treated as a continuous variable. Here, for presentation purposes, we divided here the spatial climatic gradient into quartiles (based on mean winter temperature) with equal numbers of transects within each. Lines shown are means for each of the 36 survey regions (averaged over the transects within each). Electronic supplementary material, figure S2 shows the time-series for the other climatic variables and electronic supplementary material, figure S3 shows the locations of each quartile. (Online version in colour.)

this figure shows the climatic gradient split into simplified quartiles, but climate was treated as a continuous variable in all analyses). Our 11-year dataset spanned *ca* three rodent cycles. The probability of rodent occurrence increased along the climatic harshness gradient, driven by higher occurrence probability in peak years within colder regions (electronic supplementary material, table S1). For instance, in the first peak rodent years, during 2010/2011, mean rodent occurrence across transects was 38% in the region of the warmest quantile (based on winter temperature) but 64% in the coldest. In the second peak rodent years, during 2014/2015, rodent occurrence was 18% in the warmest and 37% in the coldest.

## (c) Mean effects of climatic and rodent fluctuations on ptarmigan populations

Ptarmigan growth was higher in more climatically benign regions (positive spatial effects of climate shown in figure 5), described by warmer springs and winters, and later winters and earlier springs. Similarly, temporally, ptarmigan growth increased in years with warmer spring temperatures but was less affected by the timing of seasonal onsets and winter temperatures (temporal effects of climate shown in figure 5). Ptarmigan density was positively associated with rodent occurrence of the same year, but negatively associated with rodent occurrence in

the previous year (figure 5). Effect sizes of rodents were larger than the effect sizes of the temporal variability in climate.

## (d) Changes in the effects of climatic and rodent fluctuations along the climatic harshness gradient

The effects of rodent occurrence probability on ptarmigan growth was more positive in more climatically harsh areas compared with in more climatically benign areas (figure 6, left). In fact, the predicted effects of rodent occurrence were close to zero in regions with the warmest and earliest springs but strongly positive in the coldest regions with earliest winters and latest springs. The effects of temporal climatic variability did not greatly vary along the climate harshness gradient (figure 6, right) and 95% credible intervals for all climate interactions overlapped zero (electronic supplementary material, figure S4).

## 4. Discussion

Understanding how fluctuations in climatic conditions and species interactions affect species' population dynamics is central to current research in population ecology. We outlined two different hypotheses about how the strength of species interactions might vary along a climatic gradient, and

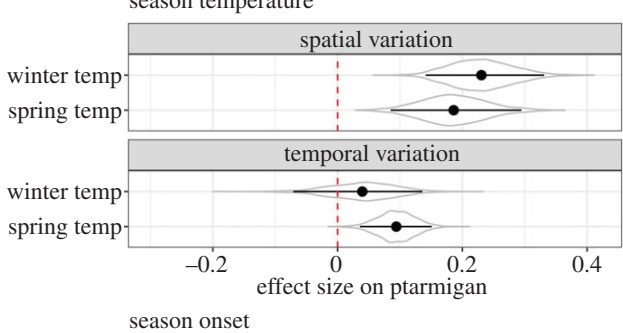

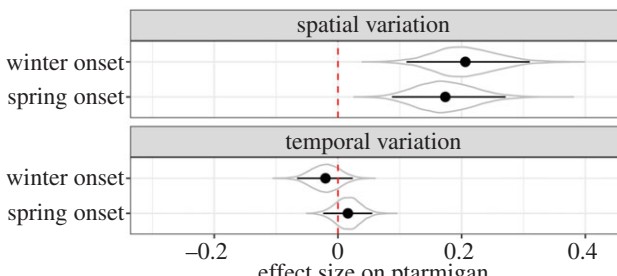

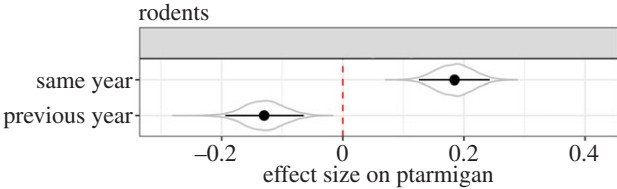

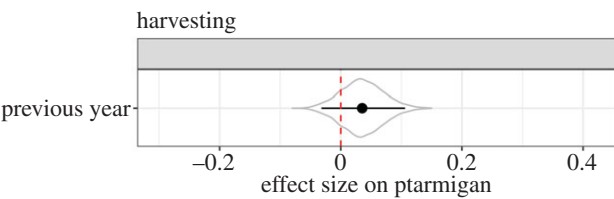

**Figure 5.** Effects of seasonal temperatures, seasonal onsets and rodent occurrence on changes in ptarmigan population density. Climate variables are split into a spatial variable (mean values across years) and a temporal variable (annual anomalies around the mean spatial variation). Covariates were standardized to units of the standard deviation to facilitate comparison of effect sizes. Violins represent the full posterior distributions. Point and ranges represent mean effects and 95% credible intervals of the distributions. The dashed vertical red line is the line of no effect. The sign of the effects of spring onset were flipped so that larger values represent lower climatic harshness (i.e. earlier springs), consistent with the other climatic variables. (Online version in colour.)

tested our predictions using a large-scale dataset on the willow ptarmigan. Our results suggest that the impacts of predator-mediated interactions on willow ptarmigan increase along the climate gradient, indicated by more strongly positive impacts of rodent dynamics in colder areas. Hence, our results were consistent with predictions stemming from the APH rather than the classical view of how species interactions change with increasing climatic harshness.

Our results support our main prediction based on the APH: a stronger link between rodent fluctuations and ptarmigan dynamics in colder regions. According to the APH, shared predators are expected to prey-switch towards rodents and away from ptarmigan, when rodents are more abundant [15,19,23]. Ptarmigan had higher growth rates during years with more rodents, which would be consistent with lower predation pressure. Hence, our results are consistent with the APH, previously tested over smaller-scales [23,25,36].

While the APH primarily considers the immediate functional response of predators, the contrasting negative lagged effects of rodents is consistent with a numerical response of predators due to greater prey density, leading to negative effects of rodents on ptarmigan in the subsequent year. The weaker effect of rodents, and hence predator-mediated interactions, on ptarmigan density in warmer regions could be caused by greater prey diversity and more generalist predators, weakening the trophic links among the dynamics of predators and prey species [8,15].

Previous research has shown how the role of species interactions can increase or decrease in strength along an ecological gradient due to different biotic and abiotic processes [5,37–39]. Probably the most common reason for changing strengths of interactions over a geographical range is changes in community assemblages and the abundance of interacting partners [3,40,41]. Our results appear to be at odds with the classic prediction of stronger species interactions at the warmer end of a species range [3], but the prediction arises in part from the assumption that the abundance of interacting partners is greater in warmer regions. The assumption may hold in many scenarios, but in our case rodents showed greater peaks in occurrence in colder regions. Indeed, other studies have shown an increase in the amplitude of rodent dynamics from southern to northern Fennoscandia [8]. Hence, our results are consistent with species interactions being important in areas where potential interacting partners are more common, at least periodically.

Another factor likely to affect the changing importance of species interactions is whether the predators are specialist or generalist species. When interactions involve specialists, they can be important over the whole distributional range of an organism [42]. Although none of the interactions between ptarmigan, rodents and predators are intrinsically specialist, they may be regarded as functionally specialist in cold regions due to the relatively low number of species in the food web. Since interaction networks at high latitudes tend to be often more specialized [43], other types of indirect interactions may also become more important in more northerly regions.

We found much weaker support for interannual climatic variability having a greater effect on ptarmigan populations in colder regions. Climatic effects may be direct and related to species' physiology, or more likely indirect and related to climate-caused changes in the availability of resources or suitable habitat. Harsh conditions may expose an organism to the edge of its niche space, hence climatic variation at the colder edge of a species' range should be more likely to affect its population growth [44]. However, the ptarmigan is a well-adapted alpine species able to inhabit regions with harsher climates than those in our study area (e.g. in Alaska and Siberia). Our study does not, therefore, address the geographical range limits of the ptarmigan. However, other studies have found clearer evidence that species' sensitivity to climate increases towards the range edge [45,46]. Further work might consider the nonlinearity in the relationship between climatic conditions and ptarmigan density to examine in more depth the effects of climatic variation at the climatic extremes.

High-altitude or -latitude organisms are among those most strongly exposed to climate change [47]. Our findings help understand the main pathways through which climate change may affect the population dynamics of ptarmigan, including the direct effects of warming versus indirect effects mediated through predators [48,49]. Ptarmigan benefit from

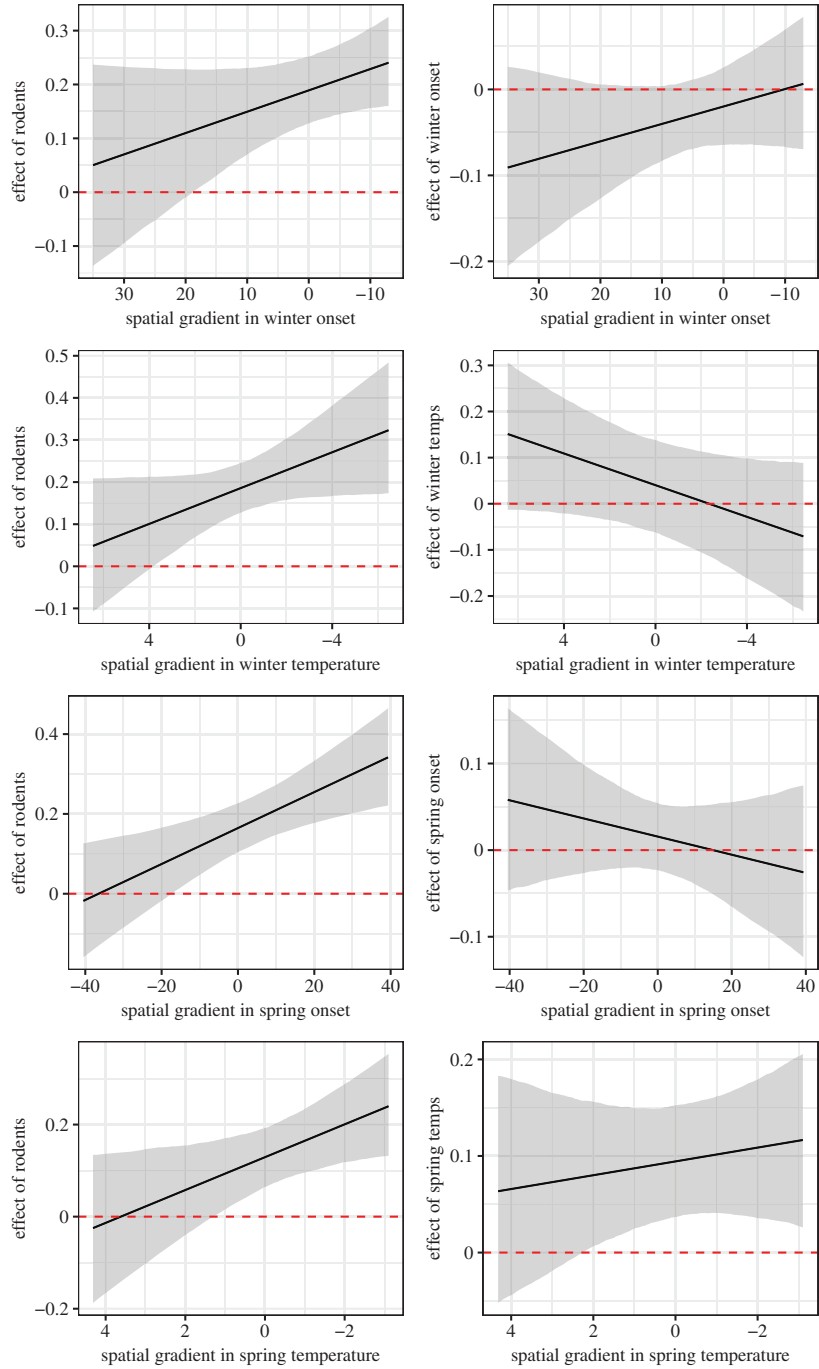

**Figure 6.** Changes in the effect sizes of temporal variability in climate and rodent occurrence and along each climatic spatial gradient on ptarmigan density. *x*-Axes are transformed so that left to right is an increasing climatic harshness gradient (i.e. the values towards the right represent colder temperatures, earlier winters and later spring). Shown are the mean standardized effect size and 95% credible interval. Each *x*-axis is centred around the median values for each variable (°C for temperatures and days for onset). Ninety-five per cent credible intervals for the interaction terms are shown in electronic supplementary material, figure S4. (Online version in colour.)

warmer springs, hence, climate change, at least in the short-term, may have some positive effects [50]. However, our results suggest that changes in predation pressure mediated via rodents are likely to be more important. One predicted conse-quence of climate change is a dampening of rodent cycles [16,51]. Dampening cycles could mean no or less frequent years of high rodent abundance, which offer temporal refuges from predation that yield 'boom' years with high ptarmigan productivity. Hence, a warming climate may lead to a more constant rate of predation pressure on ptarmigan, lowering mean population growth rates. However, dampening of the cycles might also reduce the long-term abundance of predators, lowering predation pressure on the ptarmigan [12]. In

particular, more frequent years with low vole abundance is expected to directly affect predator reproduction in the spring [51].

Our analysis has several limitations. First, the rodent data were only collected as a binary presence/absence variable along each line-transect and hence may not reflect the full changes in the relative abundances of rodents among years. Moreover, our rodent data were collected in the autumn, but rodent abundance in spring and early summer is probably the key parameter that drives predation pressure on ptarmigan and other ground-nesting birds. However, no comparable spring data for rodents are available spanning the range of our study system. Also, we could expect spring and autumn

rodent densities of the same year to be correlated [16] but that might vary with climatic conditions [51]. Overall, the patterns in the proportion of rodents seen within our data showed the expected large-scale synchronized annual cycles of rodents reported elsewhere [52]. Second, although we can reasonably infer the interaction between ptarmigan and rodents to be caused by shared predators, data on predators would also help better understand the links between predator abundance, prey preference, and alternative prey availability [23]. Last, our analysis only focused on changes in the strength of one specific type of species interaction. Willow ptarmigan may be involved in various other types of species interactions [53] such as competition with other herbivorous species, including ungulates, which differ in their strength along the climate harshness gradient. Hence, cumulatively, species interactions could still be more important at the warmer edge of its distribution.

Our analyses suggest that predator-mediated interactions become even more important in the colder regions of boreal ecosystems, contrary to the classic view that species interactions are more important at the warmer edge of species' distributions. The role of predator-mediated interactions along a climatic gradient are more generally likely to depend on the diet breadth of predators and the availability of different prey types. Rodent cycles—regarded as the heartbeat of boreal ecosystems—cause changes in prey availability that lead to predator-mediated interactions for alternative prey species. Long-term dampening of the rodent cycles that is predicted to arise due to climate change is likely to have widespread repercussions for the dynamics of many species in the boreal, especially ground-nesting birds such as the willow ptarmigan.

Data accessibility. The ptarmigan dataset (for different geographical regions) is freely available via GBIF [32,54,55]. The subset of data used for the analysis of this paper, along with the rodent and climate data, are available from the Dryad Digital Repository: https://doi.org/10.5061/dryad.tx95x69w9 [56].

Authors' contributions. E.B.N. and H.C.P. designed the line-transect sampling protocols and helped to develop and maintain the Hønsefuglportalen system. D.E.B. developed the research questions, with contributions from E.B.N. D.E.B. conducted the statistical analyses. D.E.B. wrote the first draft of the manuscript and all co-authors helped to refine the hypotheses and develop the manuscript.

Competing interests. We declare we have no competing interests

Acknowledgements. We thank Eike Lena Neuschulz, Nigel Yoccoz, Jens Åström and anonymous referees for comments on an earlier version. The work was supported by the Research Council of Norway (grant no. 251112) and by base funding from the Norwegian Institute for Nature Research. We are also grateful to all of the participants and land owners involved in the line-transect survey programme coordinated by the Rypeforvalningsprosjektet (2006–2011) and Hønsefuglportalen (from 2013–), and to the Norwegian Environment Agency for providing base funding to the programme. All surveys that were included in this study were conducted on the lands of Finnmarkseiendommen (FeFo), Statskog and Fjellstyrene, and we are particularly grateful to them for good collaboration.

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
