## [Reviewer comments · Proceedings of the Royal Society B: Biological Sciences]

Review History

RSPB-2020-0211.R0 (Original submission)

Review form: Reviewer 1

Recommendation

Major revision is needed (please make suggestions in comments)

Scientific importance: Is the manuscript an original and important contribution to its field?

Excellent

General interest: Is the paper of sufficient general interest?

Excellent

Quality of the paper: Is the overall quality of the paper suitable?

Good

Is the length of the paper justified?

Yes

Should the paper be seen by a specialist statistical reviewer?

No

Do you have any concerns about statistical analyses in this paper? If so, please specify them explicitly in your report.

Yes

It is a condition of publication that authors make their supporting data, code and materials available - either as supplementary material or hosted in an external repository. Please rate, if applicable, the supporting data on the following criteria.

Is it accessible?

Yes

Is it clear?

Yes

Is it adequate?

Yes

Do you have any ethical concerns with this paper?

No

Comments to the Author

This paper contains potentially important results by showing that spatio-temporal variation in the dynamics of ptarmigan populations is more strongly affected by rodent abundances and weather under poorer conditions. This is one of the few existing studies showing that large-scale patterns in population dynamics can be predicted by identifying some basic ecological gradients. However, I have some concerns about the analyses:

1. The model assumes a Gompertz type of density regulation but no evidence is provided for this rather strong assumption. Studies on density regulation of grouse populations should be cited to justify this assumption.
2. The authors model fluctuations in N rather than annual differences in N . It is well known in population modelling that uncertainties in the population estimates introduce spurious autocorrelations in the N -process. The analyses must be redone using ΔN as dependent variable.
3. It is unclear to me how the random effect of regions affects $\log(\sigma)$ in the observation model. Does this effect affect the shape of the curve? This must be much better explained.
4. I think that ptarmigan is a heavily hunted species in most parts of Norway. It is therefore surprising that there is no harvest term in the population model. The effects of harvesting are not even mentioned in the discussion. So this leads to the obvious question: can the stronger impact on the ptarmigan dynamics of weather/rodents under harsh conditions simply be due to reduced/increased hunting pressure in these areas?

Review form: Reviewer 2 (Nigel Yoccoz)

Recommendation

Major revision is needed (please make suggestions in comments)

Scientific importance: Is the manuscript an original and important contribution to its field?

Good

General interest: Is the paper of sufficient general interest?

Good

Quality of the paper: Is the overall quality of the paper suitable?

Acceptable

Is the length of the paper justified?

Yes

Should the paper be seen by a specialist statistical reviewer?

No

Do you have any concerns about statistical analyses in this paper? If so, please specify them explicitly in your report.

Yes

It is a condition of publication that authors make their supporting data, code and materials available - either as supplementary material or hosted in an external repository. Please rate, if applicable, the supporting data on the following criteria.

Is it accessible?

Yes

Is it clear?

No

Is it adequate?

No

Do you have any ethical concerns with this paper?

No

Comments to the Author

This paper uses 11 years (2007-2017) of ptarmigan monitoring data to assess two hypotheses: 1) that abiotic (climatic) conditions determine range limits in harsh environments, versus 2) alternative prey hypothesis, with a corollary that stronger links between prey (primary prey=rodents, secondary prey=grouse) and predators in northern environments should lead to a stronger impact of predator-prey indirect effects in northern (=harsh) environments. They used a state-space modelling approach to assess how climate and rodents dynamics influenced grouse dynamics.

My main concern is with the framing of the hypotheses and the claim that willow grouse experience harsh environments in Norway. Willow grouse occur at high densities in ecosystems with much colder spring/winter temperature, such as eastern Siberia (Kolyma; Andreev 1988) or northern Alaska/Canada (eg Tape et al. 2010). The lower densities of willow grouse in Northern Norway may be more related to the lower abundance of shrubs - we documented for example higher abundance in western Arctic Russia (which has more willow shrubs but much colder winter temperatures) than in north Norway despite what you would call "harsh" environments in north Norway. In fact, except when considering altitudinal effect (which you do not in this paper), very few of your sites can be considered close to the edge of the distribution of willow grouse, except in what you call benign environments, but I guess you do not want to discuss that edge (i.e. what determines the "warm" edge of willow grouse).

It would help to have maps of the different regions defined by your climatic gradients (for example for Figure 3, how look like the four regions delimited by the different quantiles of the climatic variables). Are the regions representing relatively contiguous areas or disjunct? Adding also the spatial units that are used in the model would help, you can easily add those in the SM. It would also help to have plots showing the data for the most important "interactions" in your models. As far as I could understand, your interactions are in fact products (i.e., what appear as CS CT and CS R on line 227 is the product of CS and CT and the product of CS and R), so they

assume linear relationships with a varying slope as shown in Figure 5. But we have no idea if the linear relationships hold. Generally you need to provide detailed diagnostics for the model – not just of the convergence of the model but of the actual relationships assumed in the model.

I was rather surprised by the model used to model rodent occurrence. First nearly all models used for rodent dynamics have been AR2 or similar structure, because of the cyclic nature of rodent dynamics (for example Bjørnstad et al. 1995 that you cite, see Cornulier et al. for a model using occurrence data). Second, why use Y_t as a predictor – i.e. is there any evidence that you have varying trends in rodent abundance over the 11 years of data after taking into account the AR2 dependency? Third, there is a large spatial variation in cycle periodicity/amplitude (see next paragraph), so that your model of rodent dynamics should consider this (ie the famous gradient in direct/delayed density-dependence, works by Bjørnstad and many others).

You need also to consider that the main impact of rodents is in the spring (when grouse breed) and not in the fall (when you sample grouse populations). There is good evidence that seasonal dynamics have changed with lower densities in spring despite smaller changes in autumn densities (see Cornulier et al. 2013), and such differences may be particularly strong when you compare “benign” and “harsh” environments. Because your benign environments may have less snow, spring densities may in fact be lower whereas fall densities are higher (see below Hansson and Henttonen 1985). This possible difference in the validity of the proxy should at least be acknowledged and assessed with existing time-series. Are those data part of the grouse data set available on GBIF? I could not find them...

L. 73 ff: the choice of references and the presentation reflect that the authors have not been following the (long...) discussion of what can cause small mammals cycles in Fennoscandia and the geographical variation in period length and amplitude. Hansson and Henttonen (1988) were the best (and first) proponents of the gradient of generalist vs specialist predators (with specialist dominating in the north and generalists dominating in the south), and the importance of snow cover (see Hansson and Henttonen 1985 for the latter). These effects were modelled by Hanski later on (Hanski et al. 1991, 1992). This is different from the indirect impact on ground-nesting birds. You for example do not mention specialist predators such as mustelids (weasels and stoats) which are both seen as THE driving force behind the rodent cycles and which may be additional predators of ground-nesting birds (especially stoats, weasels are likely true specialists, see Hellstedt et al. 2006 for a discussion).

l. 253: “highly mobile predators not likely affected by fine-scale variation”. What are the highly mobile predators? Red fox? Owls/Buzzards? Red fox are of course mobile but they are not usually put in the “highly mobile” categories, at least at a large scale (young can disperse over large distances, but they cannot sample large areas like birds of prey can do). You need to be precise when you define predators (see comments above).

Figure 3: It seems that you have missed the 2007 peak in what I suppose is partly northern Norway (“coldest”) (see Ims et al. 2011; Ehrich et al. 2020 Figure 1 and S1). Your rodent occurrence data seem to work reasonably well most of the time to identify peaks, but it would be nice to have some validation with existing time series of small rodents (e.g. TOV; Framstad 2017). And be aware that synchrony may be relatively low in regions such as Troms (contrast inner Troms in Strann et al. 2002 to Kilpisjärvi in Ehrich et al. 2020), so even if assuming synchrony may work well as a first approximation, it may have to be assessed in different regions.

Details: l. 47-48: This is a very Fennoscandian point of view – rodent population dynamics do not dominate northern boreal ecosystems in North America (snowshoe hares do – eg Krebs 2011).

l. 52: not all species continue to breed. Lemmings do, *Microtus* can, but *Myodes* spp. don't breed in winter (and they dominate in for example the boreal ecosystems=birch forests of northern Norway; Hansen et al. 1999; Ims et al. 2011).

l. 57: *Vulpes vulpes*

Nigel G. Yoccoz

Add refs:

Andreev, A. (1988). The ten year cycle of the willow grouse of lower Kolyma. *Oecologia*, 76, 261-267.

Cornulier, T., Yoccoz, N. G., Bretagnolle, V., Brommer, J. E., Butet, A., Ecke, F., . . . Lambin, X.

- (2013). Europe-wide dampening of population cycles in keystone herbivores. *SCIENCE*, 340(6128), 63-66. doi:10.1126/science.1228992
- Ehrich, D., Schmidt, N. M., Gauthier, G., Alisauskas, R., Angerbjörn, A., Clark, K., . . . Solovyeva, D. V. (2020). Documenting lemming population change in the Arctic: Can we detect trends? *AMBIO*, 49(3), 786-800. doi:10.1007/s13280-019-01198-7
- Framstad, E. (red.) 2017. Terrestrisk naturovervåking i 2016: Markvegetasjon, epifytter, smågnagere og fugl. Sammenfatning av resultater. – NINA Rapport 1376. 122 s.
- Hansen, T. F., Stenseth, N. C., & Henttonen, H. (1999). Multiannual vole cycles and population regulation during long winters: An analysis of seasonal density dependence. *The American Naturalist*, 154(2), 129-139.
- Hanski, I., Hansson, L., & Henttonen, H. (1991). Specialist predators, generalist predators, and the microtine rodent cycle. *Journal of Animal Ecology*, 60, 353-367.
- Hanski, I., Turchin, P., Korpimäki, E., & Henttonen, H. (1993). Population oscillations of boreal rodents: regulation by mustelid predators leads to chaos. *Nature*, 364, 232-235.
- Hansson, L., & Henttonen, H. (1988). Rodent dynamics as community processes. *Trends in Ecology and Evolution*, 3, 195-200.
- Hansson, L., & Henttonen, H. (1985). Gradients in density variations of small rodents: the importance of latitude and snow cover. *Oecologia*, 67, 394-402.
- Hellstedt, P., Sundell, J., Helle, P., & Henttonen, H. (2006). Large-scale spatial and temporal patterns in population dynamics of the stoat, *Mustela erminea*, and the least weasel, *M. nivalis*, in Finland. *Oikos*, 115(2), 286-298.
- Ims, R. A., Yoccoz, N. G., & Killengreen, S. T. (2011). Determinants of lemming outbreaks. *Proceedings of the National Academy of Sciences*, 108(5), 1970-1974. doi:10.1073/pnas.1012714108
- Krebs, C. J. (2011). Of lemmings and snowshoe hares: the ecology of northern Canada. *Proceedings of the Royal Society B: Biological Sciences*. doi:10.1098/rspb.2010.1992
- Strann, K.-B., Yoccoz, N. G., & Ims, R. A. (2002). Is the heart of Fennoscandian rodent cycle still beating? A 14-year study of small mammals and Tengmalm's owls in northern Norway. *Ecography*, 25(1), 81-87.
- Tape, K. D., Lord, R., Marshall, H. P., & Ruess, R. W. (2010). Snow-mediated ptarmigan browsing and shrub expansion in arctic Alaska. *Ecoscience*, 17(2), 186-193. doi:10.2980/17-2-3323
- Zimmermann, N. E., Yoccoz, N. G., Edwards, T. C., Meier, E. S., Thuiller, W., Guisan, A., . . .
- Pearman, P. B. (2009). Climatic extremes improve predictions of spatial patterns of tree species. *Proceedings of the National Academy of Sciences*, 106(Supplement 2), 19723-19728. doi:10.1073/pnas.0901643106

Decision letter (RSPB-2020-0211.R0)

06-Mar-2020

Dear Dr Bowler:

I am writing to inform you that your manuscript RSPB-2020-0211 entitled "Impacts of predator-mediated interactions along a climatic gradient on the population dynamics of an alpine bird" has, in its current form, been rejected for publication in *Proceedings B*.

This action has been taken on the advice of referees, who have recommended that substantial revisions are necessary. With this in mind we would be happy to consider a resubmission, provided the comments of the referees are fully addressed. However please note that this is not a provisional acceptance.

The resubmission will be treated as a new manuscript. However, we will approach the same reviewers if they are available and it is deemed appropriate to do so by the Editor. Please note that resubmissions must be submitted within six months of the date of this email. In exceptional

circumstances, extensions may be possible if agreed with the Editorial Office. Manuscripts submitted after this date will be automatically rejected.

Sincerely,
Dr Sasha Dall
mailto:proceedingsb@royalsociety.org

Associate Editor
Comments to Author:

Two expert referees have reviewed this manuscript favorably. However, both are quite critical of the modeling exercises reported here as explained in detailed comments. A major revision of both analysis and writing would be required for further consideration by PRSB.

Reviewer(s)' Comments to Author:

Referee: 1

Comments to the Author(s)

This paper contains potentially important results by showing that spatio-temporal variation in the dynamics of ptarmigan populations is more strongly affected by rodent abundances and weather under poorer conditions. This is one of the few existing studies showing that large-scale patterns in population dynamics can be predicted by identifying some basic ecological gradients. However, I have some concerns about the analyses:

1. The model assumes a Gompertz type of density regulation but no evidence is provided for this rather strong assumption. Studies on density regulation of grouse populations should be cited to justify this assumption.
2. The authors models fluctuations in N rather than annual differences in N . It is well known in population modelling that uncertainties in the population estimates introduce spurious autocorrelations in the N -process. The analyses must be redone using ΔN as dependent variable.
3. It is unclear to me how the random effect of regions affects $\log(\sigma)$ in the observation model. Does this effect affect the shape of the curve? This must be much better explained.
4. I think that ptarmigan is a heavily hunted species in most parts of Norway. It is therefore surprising that there is no harvest term in the population model. The effects of harvesting are not even mentioned in the discussion. So this leads to the obvious question: can the stronger impact on the ptarmigan dynamics of weather/rodents under harsh conditions simply be due to reduced/increased hunting pressure in these areas?

Referee: 2

Comments to the Author(s)

This paper uses 11 years (2007-2017) of ptarmigan monitoring data to assess two hypotheses: 1) that abiotic (climatic) conditions determine range limits in harsh environments, versus 2) alternative prey hypothesis, with a corollary that stronger links between prey (primary prey=rodents, secondary prey=grouse) and predators in northern environments should lead to a stronger impact of predator-prey indirect effects in northern (=harsh) environments. They used a state-space modelling approach to assess how climate and rodent dynamics influenced grouse dynamics.

My main concern is with the framing of the hypotheses and the claim that willow grouse experience harsh environments in Norway. Willow grouse occur at high densities in ecosystems with much colder spring/winter temperature, such as eastern Siberia (Kolyma; Andreev 1988) or northern Alaska/Canada (eg Tape et al. 2010). The lower densities of willow grouse in Northern Norway may be more related to the lower abundance of shrubs – we documented for example higher abundance in western Arctic Russia (which has more willow shrubs but much colder winter temperatures) than in north Norway despite what you would call “harsh” environments in north Norway. In fact, except when considering altitudinal effect (which you do not in this paper), very few of your sites can be considered close to the edge of the distribution of willow grouse, except in what you call benign environments, but I guess you do not want to discuss that edge (i.e. what determines the “warm” edge of willow grouse).

It would help to have maps of the different regions defined by your climatic gradients (for example for Figure 3, how look like the four regions delimited by the different quantiles of the climatic variables). Are the regions representing relatively contiguous areas or disjunct? Adding also the spatial units that are used in the model would help, you can easily add those in the SM. It would also help to have plots showing the data for the most important “interactions” in your models. As far as I could understand, your interactions are in fact products (i.e., what appear as CS CT and CS R on line 227 is the product of CS and CT and the product of CS and R), so they assume linear relationships with a varying slope as shown in Figure 5. But we have no idea if the linear relationships hold. Generally you need to provide detailed diagnostics for the model – not just of the convergence of the model but of the actual relationships assumed in the model.

I was rather surprised by the model used to model rodent occurrence. First nearly all models used for rodent dynamics have been AR2 or similar structure, because of the cyclic nature of rodent dynamics (for example Bjørnstad et al. 1995 that you cite, see Cornulier et al. for a model using occurrence data). Second, why use Y_t as a predictor – i.e. is there any evidence that you have varying trends in rodent abundance over the 11 years of data after taking into account the AR2 dependency? Third, there is a large spatial variation in cycle periodicity/amplitude (see next paragraph), so that your model of rodent dynamics should consider this (ie the famous gradient in direct/delayed density-dependence, works by Bjørnstad and many others).

You need also to consider that the main impact of rodents is in the spring (when grouse breed) and not in the fall (when you sample grouse populations). There is good evidence that seasonal dynamics have changed with lower densities in spring despite smaller changes in autumn densities (see Cornulier et al. 2013), and such differences may be particularly strong when you compare “benign” and “harsh” environments. Because your benign environments may have less snow, spring densities may in fact be lower whereas fall densities are higher (see below Hansson and Henttonen 1985). This possible difference in the validity of the proxy should at least be acknowledged and assessed with existing time-series. Are those data part of the grouse data set available on GBIF? I could not find them...

L. 73 ff: the choice of references and the presentation reflect that the authors have not been following the (long...) discussion of what can cause small mammals cycles in Fennoscandia and the geographical variation in period length and amplitude. Hansson and Henttonen (1988) were the best (and first) proponents of the gradient of generalist vs specialist predators (with specialist dominating in the north and generalists dominating in the south), and the importance of snow cover (see Hansson and Henttonen 1985 for the latter). These effects were modelled by Hanski later on (Hanski et al. 1991, 1992). This is different from the indirect impact on ground-nesting

birds. You for example do not mention specialist predators such as mustelids (weasels and stoats) which are both seen as THE driving force behind the rodent cycles and which may be additional predators of ground-nesting birds (especially stoats, weasels are likely true specialists, see Hellstedt et al. 2006 for a discussion).

l. 253: “highly mobile predators not likely affected by fine-scale variation”. What are the highly mobile predators? Red fox? Owls/Buzzards? Red fox are of course mobile but they are not usually put in the “highly mobile” categories, at least at a large scale (young can disperse over large distances, but they cannot sample large areas like birds of prey can do). You need to be precise when you define predators (see comments above).

Figure 3: It seems that you have missed the 2007 peak in what I suppose is partly northern Norway (“coldest”) (see Ims et al. 2011; Ehrich et al. 2020 Figure 1 and S1). Your rodent occurrence data seem to work reasonably well most of the time to identify peaks, but it would be nice to have some validation with existing time series of small rodents (e.g. TOV; Framstad 2017). And be aware that synchrony may be relatively low in regions such as Troms (contrast inner Troms in Strann et al. 2002 to Kilpisjärvi in Ehrich et al. 2020), so even if assuming synchrony may work well as a first approximation, it may have to be assessed in different regions.

Details: l. 47-48: This is a very Fennoscandian point of view – rodent population dynamics do not dominate northern boreal ecosystems in North America (snowshoe hares do – eg Krebs 2011).

l. 52: not all species continue to breed. Lemmings do, *Microtus can.*, but *Myodes spp.* don't breed in winter (and they dominate in for example the boreal ecosystems=birch forests of northern Norway; Hansen et al. 1999; Ims et al. 2011).

l. 57: *Vulpes vulpes*

Nigel G. Yoccoz

Add refs:

Andreev, A. (1988). The ten year cycle of the willow grouse of lower Kolyma. *Oecologia*, 76, 261-267.

Cornulier, T., Yoccoz, N. G., Bretagnolle, V., Brommer, J. E., Butet, A., Ecke, F., . . . Lambin, X. (2013). Europe-wide dampening of population cycles in keystone herbivores. *SCIENCE*, 340(6128), 63-66. doi:10.1126/science.1228992

Ehrich, D., Schmidt, N. M., Gauthier, G., Alisauskas, R., Angerbjörn, A., Clark, K., . . . Solovyeva, D. V. (2020). Documenting lemming population change in the Arctic: Can we detect trends? *AMBIO*, 49(3), 786-800. doi:10.1007/s13280-019-01198-7

Framstad, E. (red.) 2017. Terrestrisk naturovervåking i 2016: Markvegetasjon, epifytter, smågnagere og fugl. Sammenfatning av resultater. – NINA Rapport 1376. 122 s.

Hansen, T. F., Stenseth, N. C., & Henttonen, H. (1999). Multiannual vole cycles and population regulation during long winters: An analysis of seasonal density dependence. *The American Naturalist*, 154(2), 129-139.

Hanski, I., Hansson, L., & Henttonen, H. (1991). Specialist predators, generalist predators, and the microtine rodent cycle. *Journal of Animal Ecology*, 60, 353-367.

Hanski, I., Turchin, P., Korpimäki, E., & Henttonen, H. (1993). Population oscillations of boreal rodents: regulation by mustelid predators leads to chaos. *Nature*, 364, 232-235.

Hansson, L., & Henttonen, H. (1988). Rodent dynamics as community processes. *Trends in Ecology and Evolution*, 3, 195-200.

Hansson, L., & Henttonen, H. (1985). Gradients in density variations of small rodents: the importance of latitude and snow cover. *Oecologia*, 67, 394-402.

Hellstedt, P., Sundell, J., Helle, P., & Henttonen, H. (2006). Large-scale spatial and temporal patterns in population dynamics of the stoat, *Mustela erminea*, and the least weasel, *M. nivalis*, in Finland. *Oikos*, 115(2), 286-298.

Ims, R. A., Yoccoz, N. G., & Killengreen, S. T. (2011). Determinants of lemming outbreaks.

Proceedings of the National Academy of Sciences, 108(5), 1970-1974. doi:10.1073/pnas.1012714108

Krebs, C. J. (2011). Of lemmings and snowshoe hares: the ecology of northern Canada.

Proceedings of the Royal Society B: Biological Sciences. doi:10.1098/rspb.2010.1992

Strann, K.-B., Yoccoz, N. G., & Ims, R. A. (2002). Is the heart of Fennoscandian rodent cycle still beating? A 14- year study of small mammals and Tengmalm's owls in northern Norway. *Ecography*, 25(1), 81-87.

Tape, K. D., Lord, R., Marshall, H. P., & Ruess, R. W. (2010). Snow-mediated ptarmigan browsing and shrub expansion in arctic Alaska. *Ecoscience*, 17(2), 186-193. doi:10.2980/17-2-3323

Zimmermann, N. E., Yoccoz, N. G., Edwards, T. C., Meier, E. S., Thuiller, W., Guisan, A., . . .

Pearman, P. B. (2009). Climatic extremes improve predictions of spatial patterns of tree species. *Proceedings of the National Academy of Sciences*, 106(Supplement 2), 19723-19728. doi:10.1073/pnas.

Author's Response to Decision Letter for (RSPB-2020-0211.R0)

See Appendix A.

RSPB-2020-2653.R0

Review form: Reviewer 1

Recommendation

Accept as is

Scientific importance: Is the manuscript an original and important contribution to its field?

Excellent

General interest: Is the paper of sufficient general interest?

Excellent

Quality of the paper: Is the overall quality of the paper suitable?

Excellent

Is the length of the paper justified?

Yes

Should the paper be seen by a specialist statistical reviewer?

No

Do you have any concerns about statistical analyses in this paper? If so, please specify them explicitly in your report.

No

It is a condition of publication that authors make their supporting data, code and materials available - either as supplementary material or hosted in an external repository. Please rate, if applicable, the supporting data on the following criteria.

Is it accessible?

Yes

Is it clear?

Yes

Is it adequate?

Yes

Do you have any ethical concerns with this paper?

No

Comments to the Author

The authors have done an excellent job in their response to my previous concerns!

Review form: Reviewer 2 (Nigel Yoccoz)

Recommendation

Accept with minor revision (please list in comments)

Scientific importance: Is the manuscript an original and important contribution to its field?

Good

General interest: Is the paper of sufficient general interest?

Good

Quality of the paper: Is the overall quality of the paper suitable?

Excellent

Is the length of the paper justified?

Yes

Should the paper be seen by a specialist statistical reviewer?

No

Do you have any concerns about statistical analyses in this paper? If so, please specify them explicitly in your report.

No

It is a condition of publication that authors make their supporting data, code and materials available - either as supplementary material or hosted in an external repository. Please rate, if applicable, the supporting data on the following criteria.

Is it accessible?

Yes

Is it clear?

Yes

Is it adequate?

Yes

Do you have any ethical concerns with this paper?

Yes

Comments to the Author

A very thorough response, and I appreciate the efforts the authors have put in their revision. I understand the difficulties associated with showing data and not just predicted values from models, particularly when models include complex interactions. The authors now justify how they have assessed their models' goodness of fit and this is fine.

I think the authors should refer to our recently published paper (Henden et al. 2020) which address some of the same issues (effects of weather and small rodents, as well as habitat changes and carcass subsidies, using a subset of the data and using local variables with better resolution, in particular for the small rodents). I usually don't ask authors to refer to papers I have co-authored but this one supports some of the conclusions reached in this paper, but also suggest other effects.

Henden, J. A., R. A. Ims, N. G. Yoccoz, E. J. Asbjørnsen, A. Stien, J. P. Mellard, T. Tveraa, F. Marolla, and J. U. Jepsen. 2020. End-user involvement to improve predictions and management of populations with complex dynamics and multiple drivers. *Ecological Applications* 30.

Decision letter (RSPB-2020-2653.R0)

20-Nov-2020

Dear Dr Bowler

I am pleased to inform you that your manuscript RSPB-2020-2653 entitled "Impacts of predator-mediated interactions along a climatic gradient on the population dynamics of an alpine bird" has been accepted for publication in *Proceedings B*.

The referee(s) have recommended publication, but also suggest some minor revisions to your manuscript. Therefore, I invite you to respond to the referee(s)' comments and revise your manuscript. Because the schedule for publication is very tight, it is a condition of publication that you submit the revised version of your manuscript within 7 days. If you do not think you will be able to meet this date please let us know.

- 1) A text file of the manuscript (doc, txt, rtf or tex), including the references, tables (including captions) and figure captions. Please remove any tracked changes from the text before submission. PDF files are not an accepted format for the "Main Document".
- 2) A separate electronic file of each figure (tiff, EPS or print-quality PDF preferred). The format should be produced directly from original creation package, or original software format. PowerPoint files are not accepted.
- 3) Electronic supplementary material: this should be contained in a separate file and where possible, all ESM should be combined into a single file. All supplementary materials accompanying an accepted article will be treated as in their final form. They will be published alongside the paper on the journal website and posted on the online figshare repository. Files on

figshare will be made available approximately one week before the accompanying article so that the supplementary material can be attributed a unique DOI.

Sincerely,

Dr Sasha Dall

Associate Editor

Comments to Author:

The authors are encouraged to consider the request of the second reviewer in preparing their final submission.

Reviewer(s)' Comments to Author:

Referee: 1

Comments to the Author(s).

The authors have done an excellent job in their response to my previous concerns!

Referee: 2

Comments to the Author(s).

A very thorough response, and I appreciate the efforts the authors have put in their revision. I understand the difficulties associated with showing data and not just predicted values from models, particularly when models include complex interactions. The authors now justify how they have assessed their models' goodness of fit and this is fine.

I think the authors should refer to our recently published paper (Henden et al. 2020) which address some of the same issues (effects of weather and small rodents, as well as habitat changes and carcass subsidies, using a subset of the data and using local variables with better resolution, in particular for the small rodents). I usually don't ask authors to refer to papers I have co-authored but this one supports some of the conclusions reached in this paper, but also suggest other effects.

Henden, J. A., R. A. Ims, N. G. Yoccoz, E. J. Asbjørnsen, A. Stien, J. P. Mellard, T. Tveraa, F. Marolla, and J. U. Jepsen. 2020. End-user involvement to improve predictions and management of populations with complex dynamics and multiple drivers. *Ecological Applications* 30.

Author's Response to Decision Letter for (RSPB-2020-2653.R0)

See Appendix B.

Decision letter (RSPB-2020-2653.R1)

27-Nov-2020

Dear Dr Bowler

I am pleased to inform you that your manuscript entitled "Impacts of predator-mediated interactions along a climatic gradient on the population dynamics of an alpine bird" has been accepted for publication in *Proceedings B*.

Open Access

Paper charges

Sincerely,

Proceedings B

Appendix A

Reviewer(s)' Comments to Author:

Referee: 1

Comments to the Author(s)

This paper contains potentially important results by showing that spatio-temporal variation in the dynamics of ptarmigan populations is more strongly affected by rodent abundances and weather under poorer conditions. This is one of the few existing studies showing that large-scale patterns in population dynamics can be predicted by identifying some basic ecological gradients.

Response: Many thanks for the nice review of our paper.

However, I have some concerns about the analyses:

1. The model assumes a Gompertz type of density regulation but no evidence is provided for this rather strong assumption. Studies on density regulation of grouse populations should be cited to justify this assumption.

Response: This is true that we assume a Gompertz model. However, it is a pretty common and general assumption used in a wide range of population dynamic studies of different species. We now make this assumption explicit and provide some supporting references (lines 229-230 in the clean copy throughout). We address the referee's concern but adding citations where the assumption has been made in several past analyses of ptarmigan dynamics:

Aanes, S., Engen, S., Saether, B.E., Willebrand, T. & Marcstrom, V. (2002) Sustainable harvesting strategies of Willow Ptarmigan in a fluctuating environment. *Ecological Applications*, 12, 281-290.

Pedersen, H.C., Steen, H., Kastdalen, L., Broseth, H., Ims, R.A., Svendsen, W. & Yoccoz, N.G. (2004) Weak compensation of harvest despite strong density-dependent growth in willow ptarmigan. *Proceedings of the Royal Society of London Series B-Biological Sciences*, 271, 381-385.

2. The authors models fluctuations in N rather than annual differences in N. It is well known in population modelling that uncertainties in the population estimates introduce spurious autocorrelations in the N-process. The analyses must be redone using delta_N as dependent variable.

Response: We have rewritten our model as follows:

$$\ln D_{i,t} = \ln D_{i,t-1} + r_{i,t}$$

where the growth rate, r , was modelled as:

$$r_{i,t} = b_0 + b_A \ln D_{i,t-1} + L_i + SR_i + Year_t + b_{CS} CS_i + b_{CT} CT_{i,t} + b_R R_{i,t} + b_{RL} R_{i,t-1} + b_h H_{i,t-1} + b_{int_CT} CS_i CT_{i,t} + b_{int_R} CS_i R_{i,t}$$

Please see lines 226-242 for the full description of the terms.

3. It is unclear to me how the random effect of regions affects log (sigma) in the observation model. Does this effect affect the shape of the curve? This must be much better explained.

Response: We now explain this more in the text. But yes, anything affecting sigma (the parameter of the half-normal distribution that describes the shape of the curve) would affect the strength of the decline of detectability with increasing distance from the line transect. In the model for sigma (on a log-scale), we included ptarmigan group size as a fixed effect (larger groups are more detectable at further distances) and random effects for line transect and reporting regions (clusters of neighbouring line transects) to account for any spatial differences, probably related to habitat/landscape features. We also tested for effects of forest cover and precipitation during the survey but found no evidence of effects (95% CI overlapped zero), so we excluded these additional covariates. We now state this clearer on lines 191-198.

4. I think that ptarmigan is a heavily hunted species in most parts of Norway. It is therefore surprising that there is no harvest term in the population model. The effects of harvesting are not even mentioned in the discussion. So this leads to the obvious question: can the stronger impact on the ptarmigan dynamics of weather/rodents under harsh conditions simply be due to reduced/increased hunting pressure in these areas?

Response: This is a very good and relevant point, that we also spend much time discussing before preparing the first version of this draft. We have now included harvest bag statistics (from Statistics Norway, www.ssb.no) as a covariate in the model. In general, harvest quotas are set proportional to the ptarmigan density estimates of the previous survey, limiting some of the potential variability in harvest mortality. There are some issues with the harvesting data that we note here and in the paper. First, harvest bag statistics are not resolved to the level of each line

transect, rather they are available at the level of municipality (an administrative unit of which there are 356 across Norway of varying size). Second, the data for the first year of our study in 2007 were not available at the level of municipality but only pooled numbers at the coarser administrative unit of the county. We imputed the missing first year values based on the relationship between county and municipal bags in the remaining ten years from 2008-2017. Given the high R^2 of this imputation model (0.95), this imputation step likely gave reasonable estimates to downscale the 2007 data to the municipality level.

After including this additional covariate in the model, we still find the same climate-rodent effects as before, and no significant effects of harvesting. Hence, this change to the analysis has not affected our conclusions, but we agree that it was relevant to include it in the model.

Since our paper is pushing the word limit, and that harvesting was not central to our question, we include details on harvesting in the SOM (see SOM A) and only briefly mention it in our main text (lines 173-176).

Referee: 2

Comments to the Author(s)

This paper uses 11 years (2007-2017) of ptarmigan monitoring data to assess two hypotheses: 1) that abiotic (climatic) conditions determine range limits in harsh environments, versus 2) alternative prey hypothesis, with a corollary that stronger links between prey (primary prey=rodents, secondary prey=grouse) and predators in northern environments should lead to a stronger impact of predator-prey indirect effects in northern (=harsh) environments. They used a state-space modelling approach to assess how climate and rodents dynamics influenced grouse dynamics.

My main concern is with the framing of the hypotheses and the claim that willow grouse experience harsh environments in Norway. Willow grouse occur at high densities in ecosystems with much colder spring/winter temperature, such as eastern Siberia (Kolyma; Andreev 1988) or northern Alaska/Canada (eg Tape et al. 2010). The lower densities of willow grouse in Northern Norway may be more related to the lower abundance of shrubs – we documented for example higher abundance in western Arctic Russia (which has more willow shrubs but much colder winter temperatures) than in north Norway despite what you would call “harsh” environments in north Norway. In fact, except when considering altitudinal effect (which you do not in this paper), very few of your sites can be considered close to the edge of the distribution of willow grouse, except in what you call benign environments, but I guess you do not want to discuss that edge (i.e. what determines the “warm” edge of willow grouse).

Response: We very much agree with this point. We have added some text to our Introduction and Discussion sections to make it clear that we don't claim to study ptarmigan in the coldest

climates that they can tolerate. Nevertheless, our study area spanned ca. 1000 km across the Norwegian mountains and a large climate gradient over which we have standardized ptarmigan data. Testing our hypothesis requires only that some sites have relatively harsher climates and others have more relatively benign climates. But the reviewer is correct that our question is not about factors affecting range limits. We have clarified these important points on lines 94-96 and lines 402-408.

It would help to have maps of the different regions defined by your climatic gradients (for example for Figure 3, how look like the four regions delimited by the different quantiles of the climatic variables). Are the regions representing relatively contiguous areas or disjunct? Adding also the spatial units that are used in the model would help, you can easily add those in the SM.

Response: Following this suggestion, we have now included a map in the main text (Fig. 2), as well as additional ones in the supplemental online materials (Figs. S1-3). Fig. 2 show the climatic gradient and the reporting regions into which the transects cluster.

In case of any misunderstanding, we note here that the quartiles (climate split by climatic harshness) were only used for presentation purposes. In the model, we used the full continuous climate variables. The simplification of climate from continuous to categorical variable was only done to allow a simpler graphic presentation of our results. An explanation for the use of quartiles is stated in the legend of Fig. 4, and also now in the main text (lines 306-307) to make this clearer.

Note that the climatic gradient over our study sites is rather complex. The warmest transects are those closest to coastal regions, mostly in central Norway. The coldest sites could therefore be found both in the uplands of southern and central Norway as well as inland in the most northerly region of Norway. We now comment on this to make this point clearer in the text (lines 283-284).

It would also help to have plots showing the data for the most important “interactions” in your models. As far as I could understand, your interactions are in fact products (i.e., what appear as CS CT and CS R on line 227 is the product of CS and CT and the product of CS and R), so they assume linear relationships with a varying slope as shown in Figure 5. But we have no idea if the linear relationships hold. Generally you need to provide detailed diagnostics for the model – not just of the convergence of the model but of the actual relationships assumed in the model.

Response: It's a valid request but its rather hard to show the data underlying the interaction for a complex hierarchical model like ours, with multiple explanatory variables, autocorrelation and latent variables. This was one reason why we show predictions of rodent and ptarmigan dynamics based on simple space-time random effects models applied to the observed data in Figure 4. These models essentially smooth over the raw data without making any assumptions about the effects of any explanatory variables or drivers of dynamics.

On the interaction: Yes, this interpretation is correct. Like in most other models, the interaction term is the product of two main effects, in our case climate and rodents. Since our model and analysis is rather complex, we prefer to keep the linearity assumption, which is widely used in ecology for a range of different relationships. We agree that future research might develop and test hypotheses about the shape of the relationship. During our initial analysis, we considered a model with a quadratic term for the spatial climatic variable but its 95% CI did not overlap zero. As reviewer 2 notes above, we aren't really at the extremes of the climate distribution for this species, and non-linear effects at the edges of the climate range might be more apparent if we had an even greater climate gradient (now mentioned on lines 404-408).

Regarding model diagnostics, we have now extended our model checking to include residual diagnostics following the recommendations of the DHARMA package. This involves simulating data from the fitted model and creating residuals based on quantiles. These checks revealed no problems. This is now discussed on lines 273-276. These plots revealed no systematic patterns in model fit.

I was rather surprised by the model used to model rodent occurrence. First nearly all models used for rodent dynamics have been AR2 or similar structure, because of the cyclic nature of rodent dynamics (for example Bjørnstad et al. 1995 that you cite, see Cornulier et al. for a model using occurrence data). Second, why use Y_t as a predictor – i.e. is there any evidence that you have varying trends in rodent abundance over the 11 years of data after taking into account the AR2 dependency? Third, there is a large spatial variation in cycle periodicity/amplitude (see next paragraph), so that your model of rodent dynamics should consider this (ie the famous gradient in direct/delayed density-dependence, works by Bjørnstad and many others).

Response: By using a random-effects model, our model described the main spatial and temporal patterns in the rodent data based on a minimum of assumptions about the underlying drivers or mechanisms. In our view, this is appropriate in our analysis since the rodent variable is only a covariate in our main model. We did not wish to treat rodent data much different to our other covariates for climate conditions. By our approach, we simply let the data speak for itself.

We would also like to clarify what might be a misunderstanding: Y_t in the rodent model is a year random intercept effect (i.e., year is a factor) and hence it does not reflect a directional trend across years in rodent occurrence (for which year would be a continuous variable) – hence, we make no assumptions about any directional trend. We include year, survey region and year/survey regions as random effects to simply describe the spatiotemporal patterns in the data. We have now clarified this in the text (line 251).

However, we now have developed an ar2 model as the reviewer suggested to see how it compares to our random-effects model. Like our random-effects model, we yield a similar pattern of cycles with the ar2 model (shown in SOM B), with peaks in the same years. Since the results were similar between the approaches, we have retained the random-effects model as a more flexible, simpler and assumption-free framework, which we think is appropriate for our analysis since ptarmigan density was the main variable we wished to model. Moreover, when

we included this ar2 rodent model as the submodel within the ptarmigan model, we obtained the same results (SOM B), i.e., we find the ptarmigan abundance is positively linked with rodent occurrence of the same year and that the rodent effect increases with increasing climatic harshness. We now mention details of this ar2 model and alternative outputs with this model in the SOM, since there isn't space in the main text (we are at the word count) (lines 260-262 and SOM B).

You need also to consider that the main impact of rodents is in the spring (when grouse breed) and not in the fall (when you sample grouse populations). There is good evidence that seasonal dynamics have changed with lower densities in spring despite smaller changes in autumn densities (see Cornulier et al. 2013), and such differences may be particularly strong when you compare "benign" and "harsh" environments. Because your benign environments may have less snow, spring densities may in fact be lower whereas fall densities are higher (see below Hansson and Henttonen 1985). This possible difference in the validity of the proxy should at least be acknowledged and assessed with existing time-series.

Response: We now mention these important points on lines 424-430 where we recognize the limitations to our rodent data. We discuss our cross-checking with other rodent data in response to another comment below.

Are those data part of the grouse data set available on GBIF? I could not find them...

Response: As the referee noticed, the rodent data are not yet available through the data set published through GBIF. We plan to publish them to GBIF later this year; however, there are some issues regarding how to structure data sets like this one based on the Darwin Core standard, so that we want to make some community agreement on before publishing them. For the purpose of this manuscript, we will archive the rodent data used here in Dryad.

L. 73 ff: the choice of references and the presentation reflect that the authors have not been following the (long...) discussion of what can cause small mammals cycles in Fennoscandia and the geographical variation in period length and amplitude. Hansson and Henttonen (1988) were the best (and first) proponents of the gradient of generalist vs specialist predators (with specialist dominating in the north and generalists dominating in the south), and the importance of snow cover (see Hansson and Henttonen 1985 for the latter). These effects were modelled by Hanski later on (Hanski et al. 1991, 1992). This is different from the indirect impact on ground-nesting birds. You for example do not mention specialist predators such as mustelids (weasels and stoats) which are both seen as THE driving force behind the rodent cycles and which may be additional predators of ground-nesting birds (especially stoats, weasels are likely true specialists, see Hellstedt et al. 2006 for a discussion).

Response: We have now revised this section of the paper, including the additional references suggested by the reviewer. But we still rather keep it brief since, as the reviewer notes, this discussion can be found already elsewhere and our paper is already at the word limit.

I. 253: “highly mobile predators not likely affected by fine-scale variation”. What are the highly mobile predators? Red fox? Owls/Buzzards? Red fox are of course mobile but they are not usually put in the “highly mobile” categories, at least at a large scale (young can disperse over large distances, but they cannot sample large areas like birds of prey can do). You need to be precise when you define predators (see comments above).

Response: We have revised this sentence to compare movements of raptors and mammals.

Figure 3: It seems that you have missed the 2007 peak in what I suppose is partly northern Norway (“coldest”) (see Ims et al. 2011; Ehrich et al. 2020 Figure 1 and S1). Your rodent occurrence data seem to work reasonably well most of the time to identify peaks, but it would be nice to have some validation with existing time series of small rodents (e.g. TOV; Framstad 2017). And be aware that synchrony may be relatively low in regions such as Troms (contrast inner Troms in Strann et al. 2002 to Kilpisjärvi in Ehrich et al. 2020), so even if assuming synchrony may work well as a first approximation, it may have to be assessed in different regions.

Response: We additionally compiled rodent monitoring data from the terrestrial monitoring program (TOV-E) coordinated by Erik Framstad. The report on these data is reference no. 52 in our paper. These field data are collected with snap traps or live traps and provide an abundance index. These datasets are collected in different locations of Norway to our ptarmigan data. However, despite the differences in areas and study protocols, our rodent data captures the peak and trough years shown by these other rodent data. Below shows a plot of data from the TOV-E program (left) and the rodent data in our analysis (right) – they are aligned by county administration regions in Norway but they are collected at different study sites within each region. Hence, some differences between the programs would be expected based on local variation in rodent numbers. While our data has some limitations (primarily that it is rodent occurrence and not abundance), it has the advantage of being directly linked to our study sites, so we feel it is the best data source for our question. Using the TOV data shown above would require making strong assumptions about the spatial pattern of rodent dynamics to match it with our ptarmigan data. We state the limitation of our study with regards to the rodent data (lines 422-430).

Figure showing the alternative rodent data from the TOV-E program (left) and our data(right). The panels show different counties but we note that the datasets were collected at different locations within each.

Details: l. 47-48: This is a very Fennoscandian point of view – rodent population dynamics do not dominate northern boreal ecosystems in North America (snowshoe hares do – eg Krebs 2011).

Response: We have revised this sentence (lines 49-50).

l. 52: not all species continue to breed. Lemmings do, *Microtus can*, but *Myodes spp.* don't breed in winter (and they dominate in for example the boreal ecosystems=birch forests of northern Norway; Hansen et al. 1999; Ims et al. 2011).

Response: This sentence has been deleted.

l. 57: *Vulpes vulpes*

Response: Corrected.

Nigel G. Yoccoz

Response: Thank you for the suggestions. We have now added many of these to our paper but since we are close to the word limit, we cannot add them all.

Add refs:

Andreev, A. (1988). The ten year cycle of the willow grouse of lower Kolyma. *Oecologia*, 76, 261-267.

Cornulier, T., Yoccoz, N. G., Bretagnolle, V., Brommer, J. E., Butet, A., Ecke, F., . . . Lambin, X. (2013). Europe-wide dampening of population cycles in keystone herbivores. *SCIENCE*, 340(6128), 63-66. doi:10.1126/science.1228992

Ehrich, D., Schmidt, N. M., Gauthier, G., Alisauskas, R., Angerbjörn, A., Clark, K., . . . Solovyeva, D. V. (2020). Documenting lemming population change in the Arctic: Can we detect trends? *AMBIO*, 49(3), 786-800. doi:10.1007/s13280-019-01198-7

Framstad, E. (red.) 2017. Terrestrisk naturovervåking i 2016: Markvegetasjon, epifytter, smågnagere og fugl. Sammenfatning av resultater. – NINA Rapport 1376. 122 s.

Hansen, T. F., Stenseth, N. C., & Henttonen, H. (1999). Multiannual vole cycles and population regulation during long winters: An analysis of seasonal density dependence. *The American Naturalist*, 154(2), 129-139.

Hanski, I., Hansson, L., & Henttonen, H. (1991). Specialist predators, generalist predators, and the microtine rodent cycle. *Journal of Animal Ecology*, 60, 353-367.

Hanski, I., Turchin, P., Korpimäki, E., & Henttonen, H. (1993). Population oscillations of boreal rodents: regulation by mustelid predators leads to chaos. *Nature*, 364, 232-235.

Hansson, L., & Henttonen, H. (1988). Rodent dynamics as community processes. *Trends in Ecology and Evolution*, 3, 195-200.

Hansson, L., & Henttonen, H. (1985). Gradients in density variations of small rodents: the importance of latitude and snow cover. *Oecologia*, 67, 394-402.

Hellstedt, P., Sundell, J., Helle, P., & Henttonen, H. (2006). Large-scale spatial and temporal patterns in population dynamics of the stoat, *Mustela erminea*, and the least weasel, *M-nivalis*, in Finland. *Oikos*, 115(2), 286-298.

Ims, R. A., Yoccoz, N. G., & Killengreen, S. T. (2011). Determinants of lemming outbreaks. Proceedings of the National Academy of Sciences, 108(5), 1970-1974. doi:10.1073/pnas.1012714108

Krebs, C. J. (2011). Of lemmings and snowshoe hares: the ecology of northern Canada. Proceedings of the Royal Society B: Biological Sciences. doi:10.1098/rspb.2010.1992

Strann, K.-B., Yoccoz, N. G., & Ims, R. A. (2002). Is the heart of Fennoscandian rodent cycle still beating? A 14- year study of small mammals and Tengmalm's owls in northern Norway. Ecography, 25(1), 81-87.

Tape, K. D., Lord, R., Marshall, H. P., & Ruess, R. W. (2010). Snow-mediated ptarmigan browsing and shrub expansion in arctic Alaska. *Ecoscience*, 17(2), 186-193. doi:10.2980/17-2-3323

Zimmermann, N. E., Yoccoz, N. G., Edwards, T. C., Meier, E. S., Thuiller, W., Guisan, A., . . . Pearman, P. B. (2009). Climatic extremes improve predictions of spatial patterns of tree species. *Proceedings of the National Academy of Sciences*, 106(Supplement 2), 19723-19728. doi:10.1073/pnas.0901643106

Appendix B

:The authors are encouraged to consider the request of the second reviewer in preparing their final submission.

Reviewer(s)' Comments to Author:

Referee: 1

Comments to the Author(s).

The authors have done an excellent job in their response to my previous concerns!

Response: Thank you!

Referee: 2

Comments to the Author(s).

A very thorough response, and I appreciate the efforts the authors have put in their revision. I understand the difficulties associated with showing data and not just predicted values from models, particularly when models include complex interactions. The authors now justify how they have assessed their models' goodness of fit and this is fine.

I think the authors should refer to our recently published paper (Henden et al. 2020) which address some of the same issues (effects of weather and small rodents, as well as habitat changes and carcass subsidies, using a subset of the data and using local variables with better resolution, in particular for the small rodents). I usually don't ask authors to refer to papers I have co-authored but this one supports some of the conclusions reached in this paper, but also suggest other effects.

Henden, J. A., R. A. Ims, N. G. Yoccoz, E. J. Asbjørnsen, A. Stien, J. P. Mellard, T. Tveraa, F. Marolla, and J. U. Jepsen. 2020. End-user involvement to improve predictions and management of populations with complex dynamics and multiple drivers. *Ecological Applications* 30.

Response: Thank you - we have now added this reference to the Discussion to refer to other factors affecting ptarmigan density (reference no. 53).